# Development and Characterization of a Novel Peptide—Drug Conjugate with DM1 for Treatment of FGFR2-Positive Tumors

**DOI:** 10.3390/biomedicines9080849

**Published:** 2021-07-21

**Authors:** Yayu Wang, Yadan Li, Jieqiong Cao, Qilin Meng, Xiaocen Li, Yibo Zhang, Kit S. Lam, An Hong, Ruiwu Liu, Xiaojia Chen

**Affiliations:** 1Department of Cell Biology, Institute of Biomedicine, Jinan University, Guangzhou 510000, China; wangyayu@trinomab.com (Y.W.); yadanli357@gmail.com (Y.L.); caojieqiong1993@gmail.com (J.C.); mengqilin365@gmail.com (Q.M.); yibozhang@jnu.edu.cn (Y.Z.); tha@jnu.edu.cn (A.H.); 2National Engineering Research Center of Genetic Medicine, Guangdong Provincial Key Laboratory of Bioengineering Medicine, Guangzhou 510000, China; 3Guangdong Provincial Engineering Research Center of Biotechnology, Guangzhou 510000, China; 4UC Davis NCI-Designated Comprehensive Cancer Center, Department of Biochemistry and Molecular Medicine, University of California Davis, Sacramento, CA 95817, USA; lixiaocenyesyes@gmail.com (X.L.); Kit.lam@ucdmc.ucdavis.edu (K.S.L.)

**Keywords:** peptide–drug conjugates (PDCs), one-bead one-compound (OBOC), peptide screening, FGFR2, cancer, targeted therapy

## Abstract

A maytansin derivative, DM1, is a promising therapeutic compound for treating tumors, but is also a highly poisonous substance with various side effects. For clinical expansion, we tried to develop novel peptide–drug conjugates (PDCs) with DM1. In the study, a one-bead one-compound (OBOC) platform was used to screen and identify a novel, highly stable, non-natural amino acid peptide targeting the tyrosine receptor FGFR2. Then, the identified peptide, named LLC2B, was conjugated with the cytotoxin DM1. Our results show that LLC2B has high affinity for the FGFR2 protein according to an isothermal titration calorimetry (ITC) test. LLC2B-Cy5.5 binding to FGFR2-positive cancer cells was confirmed by fluorescent microscopic imaging and flow cytometry in vitro. Using xenografted nude mouse models established with breast cancer MCF-7 cells and esophageal squamous cell carcinoma KYSE180 cells, respectively, LLC2B-Cy5.5 was observed to specifically target tumor tissues 24 h after tail vein injection. Incubation assays, both in aqueous solution at room temperature and in human plasma at 37 °C, suggested that LLC2B has high stability and strong anti-proteolytic ability. Then, we used two different linkers, one of molecular disulfide bonds and another of a maleimide group, to couple LLC2B to the toxin DM1. The novel peptide–drug conjugates (PDCs) inhibited tumor growth and significantly increased the maximum tolerated dose of DM1 in xenografted mice. In brief, our results suggest that LLC2B–DM1 can be developed into a potential PDC for tumor treatment in the future.

## 1. Introduction

Fibroblast growth factor receptors (FGFRs) are important members of the tyrosine kinase receptor family [1]. The FGFR family contains four members: FGFR1, FGFR2, FGFR3 and FGFR4 [2]. Most of them are single-chain glycoprotein molecules, which are divided into the extracellular region, transmembrane region and intracellular region [3]. Under normal physiological conditions, an FGFR binds to its ligand fibroblast growth factor (FGF). The FGFR dimer and self-phosphorylation activate downstream signaling pathways such as the JAK/STAT pathway and phospholipase C pathway, which play important roles in tumor growth and angiogenesis [4]. Abnormalities in FGFR are closely related to the occurrence of many types of tumors, such as lung cancer, breast cancer, gastric cancer, etc. Therefore, FGFR-targeting drugs can have direct or indirect antitumor effects [5,6].

Erdafitinib is the first FDA-approved FGFR tyrosine kinase inhibitor for treating urothelial carcinoma [7]. There are currently 87 FGFR-targeting drugs at the clinical stage. These drugs are mainly used in the field of tumor treatment, such as infratinib, pemigatinib, rogaratinib, AZD-4547, PRN-1371 and mastinib mesylate. Among them, seven drugs are in clinical phase III [8]. It is expected that more FGFR drugs will enter the market in the next few years. Therefore, the competition for FGFR-target inhibitors is high.

An antibody–drug conjugate (ADC) is a monoclonal antibody coupled with a cytotoxin through a specific linker, which combines the high selectivity of antibodies with an antitumor toxin. Its overall structure can be divided into three different structural modules: an antibody, cytotoxic compound and linker [9,10]. Compared with traditional chemotherapy drugs, an ADC has stronger killing effects on cancer cells. It exerts activity at pmol and nmol concentrations and should not exhibit immunogenicity after entering the human body [11,12]. Ten ADC drugs have been approved by the FDA to date. Among them, Polivy, Padcev, Enthertu and Trodelvy had just been approved in the previous year at the time of writing. This indicates a breakthrough in this development field.

Compared with ADCs, peptide–drug conjugates (PDCs) are in the development stage. Similarly, a PDC is a target peptide ligand conjugated to a cytotoxin via a cleavable linker. PDCs have many advantages, such as small size, excellent cell permeability and high drug-loading capability [13]. However, how to obtain an efficient target peptide to specifically recognize the tumor is very important for PDC development.

The OBOC combination peptide library was developed more than two decades ago, which can generate thousands to millions of resin beads containing compounds for rapidly screening a peptide, and each bead contains only one compound [14]. Compared to phage-display methods, we can obtain peptides containing unnatural amino acid residues by using the OBOC library. The obtained peptides have high stability because they are not easily recognized and degraded by proteases [15]. Moreover, the OBOC technology offers many more structural possibilities, e.g., linear, cyclic, branched and macrocyclic peptide libraries, which are typically synthesized on solid-phase resin beads using fmoc chemistry and split-mix synthesis [16].

In this study, we tried to develop novel PDCs with high affinity for FGFR2 for tumor treatment. Using OBOC technology and FGFR2-positive cells, we screened and obtained a specific peptide named LLC2B, with 8 unnatural amino acid residues. Then, we coupled the LLC2B peptide with the cytotoxin DM1 to form a peptide–drug conjugate, which has an obvious inhibitory effect on tumor proliferation in vitro and in vivo.

## 2. Materials and Methods

### 2.1. Materials

The peptides named LLC2A, LLC2B, LLC2C, LLC2D, LLC2E and LLC2F were all synthesized by Shanghai Qiangyao Biological Technology Co., Ltd., Shanghai, China. The OBOC peptide library and PDC were synthesized by the Kit Lam laboratory, UC Davis, United States. The linking molecules Fmoc-Lys (Dde)-OH and Fmoc-AEEA, which were for drug synthesis, were purchased from ChemPep Company, Palm Beach, FL, USA; amide resin was purchased from P3BioSystems Co., Ltd., Louisville, KY, USA; HCTU and 3-(2-pyridyldithio)propionic acid were purchased from Matrix Scientific Co., Ltd., Elgin, SC, USA; 6-Cl HOBt and natural and unnatural amino acids were all purchased from Chem-Impex Co., Ltd., Bensenville, IL, USA; fluorescein-labeled Cyanine5.5-NHS was purchased from LuMiprobe Co., Ltd.; 4-methylpiperidine was purchased from Alfa Aesar Co., Ltd., Ward Hill, MA, USA; DIEA, trifluoroacetic acid, anisole, DIC and other organic solvents were all purchased from Sigma-Aldrich Co., Ltd., Saint Louis, MO, USA; DM1 was purchased from BoRui Co., Ltd., SuZhou, China. FGFR2 was purchased from Sino Biological Co., Ltd., Beijing, China; anti-FGFR2 (Bek(C-17):sc-122) was acquired from Santa Cruz Blotechnology, Santa Cruz, CA, USA.

### 2.2. Cells

The HEK293, HUVEC, MCF-7, EC109, KYSE180 and KYSE510 cell lines were purchased from the American Type Culture Collection (ATCC). All the cells were cultured in RPMI-1640 Medium (Gibco, USA), which was supplemented with 10% fetal bovine serum (Biological Industrie, Israel). The cell lines were incubated in a humidified atmosphere with 5% CO_2_ at 37 °C.

### 2.3. Procedure for the Identification of FGFR2-Binding Peptides

DMF (90%) was used to suspend the OBOC library beads and place them into 6-well plates with even spreading. The plate was not moved, and the beads were allowed to attach to the bottom of the plate for 1.5–2 h. Then, the 90% DMF was discarded, and the beads were washed once with 50% DMF. Then, the beads were washed with sterile water three times and PBS twice in a clean cell culture hood. Next, 293 and 293-FGFR2 cells were collected in sterile tubes with complete medium (CM). The cell density was adjusted to 1 × 10^6^ per mL with CM. First, 1 mL of CM was added into each well with beads, and then, 500 µL of 293 cells and 500 µL of 293-FGFR2^+^ cells were added. In total, there were 1 × 10^6^ cells in 2 mL of CM per well. The plate was placed in a 37 °C, 5% CO_2_ incubator with gentle shaking for 1 h. The CM was discarded, the cells were washed with PBS, and PBS was added to cover the cells. The wells were observed under a fluorescent microscope to pick out the beads binding in the 293-FGFR2 cells with GFP expression. The selected beads were put into PBS buffer, and then, 8 M guanidine hydrochloride was used to wash away attached cells. Consequently, the peptides on the isolated beads were sequenced using an amino acid sequencer. The peptides were then synthesized according to the results of the sequencing. The sequences of the peptides are shown in Appendix A.

### 2.4. The Synthesis of LLC2B-Mal-DM1 and LLC2B-SS-DM1

First, the protected LLC2B-Lys (Dde) was assembled on rink amide resin using HCTU/DIEA activation for Fmoc/t-Bu chemistry. Then, half of the protected LLC2B-Lys (Dde) beads were transferred to a 10-mL polypropylene column with a frit. The Dde protecting group was removed with 2% NH_2_NH_2_ in DMF twice (5 min and 10 min). The beads were washed with DMF, MeOH and DMF, three times with each agent. 3-Maleimidopropionic acid (213 mg, 1.26 mmoL, 5 eq. to beads), 6-Cl HOBt (213 mg, 1.26 mmoL), and DIC (195 µL, 1.26 mmoL) were dissolved in DMF (6 mL) and mixed for 15 min before being added to the beads. The coupling reaction was conducted at room temperature for 2 h to yield protected LLC2B-K (Mal) beads. After the liquid was drained by vacuum, the beads were thoroughly washed with DMF, MeOH and dichloromethane (DCM), three times with each, and then dried under vacuum. The LLC2B-K (Mal) was cleaved off the beads with a TFA cocktail containing TFA (87.5%)/thioanisole (5%)/triisopropylsilane (2.5%)/water (5%) for 3 h. The liquid was collected and precipitated with cold ethyl ether. The precipitate was washed two more times with cold ethyl ether and dried over vacuum. LLC2B-K(Mal) (150 mg, 0.06 mmoL) powder was weighed out and dissolved in a mixture of acetonitrile (ACN) and water (1:1), and then added to PBS buffer (pH 7.2) containing 10 mM EDTA; then, through HCl and NaOH adjusted the pH to 7.0. A solution of DM1 (40 mg, 0.054 mmoL) in ACN was slowly added to the solution of LLC2B-K (Mal), which was stirred at room temperature for 1 h, until an Ellman test was negative. The resulting solution was directly used for reversed-phase HPLC purification on a preparative Vydac C18 column (22 × 250 mm) with a flow rate of 5 mL/min. The synthesis approach for LLC2B-SS-DM1 is similar to this method. We just replaced the 3-nmaleimidopropionic acid with 3-(2-pyridyldithio) propionic acid.

### 2.5. The Synthesis of LLC2B-Cy5.5 and S-LLC2B-Cy5.5

A portion of the protected-LLC2B-Lys (Dde) beads (~50 mg) was transferred to a 1.5 mL polypropylene column with a frit. After the Dde beads were removed and thoroughly wash as indicated above, Cyanine5.5-NHS (38 mg, 0.05 mmoL) and DIEA (17.5 µL, 0.1 mmoL) in anhydrous DMF (0.7 mL) were added to the beads, which were then incubated with rotation at room temperature in the dark (column wrapped with foil) for 2 h. The crude product was cleaved off the beads using a TFA cocktail as described above and precipitated with cold ethyl ether. HPLC purification yielded LLC2B-Cyanine5.5 as a dark blue powder (purity > 95%). The identity was confirmed by MALDI-TOF MS on a BRUKER microflex (ion positive reflector mode). MALDI-TOF MS Calculated: 3111.67. Found: 3112.21. The synthesis of S-LLC2B-Cyanine5.5 is similar to that of LLC2B-Cyanine5.5, except that the amino acid sequence is (hydroxyproline)-(D-Leu)-(citrulline)-(D-Asp)-(D-Asn)-(D-Lys)-(1-aminocyclopropanecarboxylic acid)-(norvaline)-(2-aminoadipic acid), the structure of amino acids is shown in Appendix A.

### 2.6. Isothermal Titration Calorimetry (ITC)

ITC experiments were carried out with an ITC200 microcalorimeter (MicroCal, Northampton, MA, USA) as described previously [17]. Both the LLC2B peptide and the recombinant proteins of the FGFR2 (Sino Biological, Beijing, China) were dissolved in sterile water. These samples were thoroughly degassed and then centrifuged to remove precipitates. The binding of both was measured by ITC at 25 °C using 0.1 mM peptide in the sample cell and 0.001 mM FGFR protein in the injecting syringe. Throughout the experiment, 19 drops were injected into the sample cell, and each drop had a volume of 2 μL. The interval between each drop was 5 min. Origin 7 software (Northampton, MA, USA) was applied for evaluating the experimental raw ITC data.

### 2.7. Immunofluorescence

For in vitro immunofluorescence analysis, cells were fixed in 4% paraformaldehyde at room temperature for 15 min and then washed 3 times for 5 min each with PBS. Subsequently, the cells were incubated for 10 min in permeabilization solution (PBS; 0.25% Triton X-100) and then washed again with PBS 3 times for 5 min each. The cells were blocked in blocking solution (PBS; 1% BSA; 0.1% Tween 20) for 30 min, incubated overnight at 4 °C with primary antibodies, anti-FGFR2 (Bek(C-17):sc-122) in blocking solution, and washed intensively 5 times for 5 min each with PBST. Cy5.5-labeled secondary antibody was then applied for 1 h at room temperature, following which the cells were stained with DAPI (staining of nuclei) for 10 min. The images were acquired on a confocal microscope (Zeiss LSM 700, Germany).

### 2.8. In Vivo Targeting of LLC2B

PBS, S-LLC2B-L-Cy5.5 (40 nmoL) and LLC2B-L-Cy5.5 (40 nmoL) were injected intravenously into MCF-7 or KYSE180 tumor-bearing nude mice. After 24 h, the Cy5.5 fluorescence in the entire body of the mouse was acquired by using an In-Vivo Xtreme (Bruker, Germany). After the mouse was sacrificed, the Cy5.5 fluorescence in the tumor, brain, liver, heart, spleen and lung tissues was quantified. The radiant efficiency was measured using the MI SE software (Bruker, Germany) and normalized by tissue volume.

### 2.9. Peptide Stability Testing In Vitro

First, 9 mL of human plasma was mixed with 1 mL of LLC2B, with a final LLC2B concentration of 100 µM, and the mixture was incubated at 37 °C. Then, 250 μL of the mixture was removed after 0, 2, 4, 6, 12, 24, 48, 72, 96, 120 and 144 h of incubation, into 3 tubes each time. Next, 100 µL of TFA added, and the mixture was vortexed for 30 s, placed at room temperature for 10–30 min, and centrifuged at 13,000× *g* rpm/min for 15 min. The supernatant was removed and centrifuged for 5 min, and then, the new supernatant was taken for analysis. The peak area method was used for quantitative detection, and the peak area was used to generate a standard curve for obtaining the polypeptide concentrations at the time points. A similar method was used to measure the stability at room temperature. The peptide was placed at room temperature without plasma incubation, and samples were taken at 0, 12, 24, 48, 72, 96 and 142 h and used for high-performance liquid chromatography (HPLC) (ThermoFly, Ultimate 3000) for quantitative analysis. The chromatographic column used was purchased from ThermoFly (Acclaim 300 C18 LC ColuMns, 4.6 mm diameter, 150 mm length, Catalog #: 060266).

A similar method was used to measure the stability of LLC2B-Mal-DM1 and LLC2B-SS-DM1 at room temperature. The peptide was placed at room temperature without plasma incubation, and samples were taken at 0, 30, 60, 120, 240, and 2880 min and used for liquid chromatography mass spectrometry (LC-MS) (Agilent 6545 LC/Q-TOF MS) for quantitative analysis.

### 2.10. Clone Formation Experiment

We seeded 200 cells per well in 6-well plates with culture medium. After 12 h, the cells were incubated with different agents for 1 week at 37 °C with 5% CO_2_. Then, we washed the wells with PBS, fixed the cells with methanol for 10 min and stained them with crystal violet at room temperature. Finally, we washed them with sterile water several times in order to obtain a clean background, and then, the colonies formed were photographed and statistically analyzed.

### 2.11. Tumor-Targeting Analysis via Flow Cytometry

FGFR2-positive cells (80–90%) were dissociated with 0.05% trypsin–EDTA, which was then neutralized with culture medium. To analyze the binding ability of the peptides, the cells (3 × 10^5^) in each sample were incubated with a 2 µL equal amount of LLC2B-Cy5.5 or S-LLC2B-Cy5.5 for 30 min on ice. Then, each sample was washed three times with 1 mL of PBS. Finally, these samples in PBS were analyzed by flow cytometry, and the mean fluorescence intensity (MFI) of each individual sample was evaluated by flow cytometry.

### 2.12. Antitumor Study In Vivo

Six-week-old female BALB/c nude mice were purchased from the Animal Experiment Center of Sun Yat-Sen University. A total of 7.0 × 10^6^ MCF-7 cells in 100 μL of PBS were subcutaneously injected into the right flanks of the mice. The tumors were allowed to grow up to 0.3 cm^3^ (volume = (length × width^2^)/2). The mice were divided into 9 groups (n = 5 mice/group): PBS, free DM1 (0.5 mg/kg), free DM1 (1.0 mg/kg), LLC2B-Mal-DM1 (1.0 mg DM1 equiv./kg), LLC2B-Mal-DM1 (2.0 mg DM1 equiv./kg), LLC2B-Mal-DM1 (4.0 mg DM1 equiv./kg), LLC2B-SS-DM1 (0.5 mg DM1 equiv./kg), LLC2B-SS-DM1 (1.0 mg DM1 equiv./kg) and LLC2B-SS-DM1 (2.0 mg DM1 equiv./kg). Each mouse was intravenously injected with solutions once a week for 3 weeks, and the tumor volume was measured once every three days for 4 weeks.

### 2.13. Statistics

All the numerical data are expressed as the mean ± S.D. Statistical differences between two groups were determined by Student’s *t* test. *p* < 0.05 was considered significant, (*) for *p* < 0.05, (**) for *p* < 0.01 and (***) for *p* < 0.001.

## 3. Results

### 3.1. Establishment of the OBOC Specific Peptide Library and the Specific Short Peptide Screening

A diagram of the OBOC specific peptide library is shown in Figure 1A. The OBOC library was synthesized on TentaGel S NH_2_ resin (0.26 mmoL/g) with Fmoc chemistry, employing a “split-mix” strategy, using 6-Cl HOBt/DIC as coupling reagents, as described in previous publications [18,19,20]. Each “X” symbol means 42 unnatural amino acids at random, and the core sites “QAE” were replaced with several unnatural amino acids of similar structure (Appendix A). The peptide library consists of millions of beads, with each bead containing a single peptide. The recombinant FGFR2-positive 293T cells with the eGFP tag were obtained by using a lentiviral expression system. Cells mixed with equal amounts of recombinant FGFR2-293T and blank 293T cells were incubated with the peptide library for 1 h. With the assistance of a fluorescent microscope, the beads binding to the FGFR2-293T cells were identified and isolated. Then, the cells were washed away from the isolated beads, and the peptides bound to the beads were sequenced using an amino acid sequencer. The peptides were then synthesized according to the results of the sequencing. These peptides could be conjugated with a drug or fluorescent marker for further study (Figure 1B).

Finally, we were able to obtain six peptides from the screening of the OBOC peptide library, which were named LLC2A to LLC2F. We then performed ITC experiments to verify the affinity of these short peptides for FGFR2. The results show that both LLC2B and LLC2C have high affinity for the FGFR2 extracellular motif protein, and that of LLC2B was much higher (Figure 1C). Therefore, LLC2B was chosen for further research. Its sequence is (hydroxyproline)–(D-Leu)–(citrulline)–(D-Asp)–(D-Asn)–(D-Lys)–(1-aminocyclopropanecarboxylic acid)–(norvaline)–(2-aminoadipic acid).

### 3.2. Targeting of FGFR2-Positive Cancer Cells by LLC2B In Vitro

In the study, the negative control peptide named S-LLC2B, (hydroxyproline)–(D-Leu)–(citrulline)–(D-Asp)–(D-Asn)–(D-Lys)–(1-aminocyclopropanecarboxylic acid)–(norvaline)–(2-aminoadipic acid), was used, which has the same amino acids but in a scrambled sequence. To verify LLC2B’s affinity in vivo and in vitro, we synthesized LLC2B-L-Cy5.5, which consists of LLC2B, a linker and fluorescent Cy5.5. Similarly, S-LLC2B-L-Cy5.5 was synthesized as a control.

To determine whether LLC2B was bound to the tumor cells, LLC2B-L-Cy5.5 or S-LLC2B-L-Cy5.5 was added to the culture medium of FGFR2-positive (KYSE180, MCF-7) and FGFR2-negative cell lines (EC109) [21,22]. After 30 min, flow cytometry showed that LLC2B-L-Cy5.5 could bind to FGFR2-positive cells more efficiently than could S-LLC2B-L-Cy5.5, but neither bound to FGFR2-negative cells (Figure 2A).

Furthermore, the recombinant cell lines EC109-FGFR2 which label GFP, overexpressing FGFR2. EC109-CTL which label GFP, with an empty vector, as for control, described for the previous study were used. LLC2B-L-Cy5.5 or S-LLC2B-L-Cy5.5 were also added to the culture medium of EC109-FGFR2 or EC109-CTL cells. As Figure 2B shows, with the antibody of FGFR2 used as a positive control, LLC2B-L-Cy5.5 binds to EC109-FGFR2 cells significantly more strongly than to EC109-CTL. S-LLC2B-L-Cy5.5 binds to EC109-FGFR2 and EC109-CTL cells with lower affinity. It was suggested that LLC2B can specifically target the FGFR2 receptor on cell membranes (Figure 2B).

### 3.3. Targeting of FGFR2-Positizve Tumor by LLC2B In Vivo

To verify whether LLC2B could target tumors in vivo, we established xenografted mice bearing KYSE180 and MCF-7 cells by subcutaneous transplantation. First, 40 nmoL of LLC2B-L-Cy5.5 or S-LLC2B-L-Cy5.5 was injected into the tail vein of a xenografted mouse, while PBS injections were given in the control group. After 24 h, every mouse was imaged by the localization of Cy5.5 in an in vivo small animal imaging device. Then, each mouse was sacrificed, and a set of organs and tumor were harvested and also imaged in a device. The ex vivo fluorescent imaging data revealed strong fluorescence signals in the tumor tissues compared to other non-cancerous tissues in the mice treated with LLC2B-L-Cy5.5.

In the MCF-7 xenografted mouse model, the fluorescence signal was mainly observed in the tumors in the LLC2B-L-Cy5.5 group, and it was much higher than in the control group with S-LLC2B-Cy5.5. The ex vivo fluorescent imaging data revealed strong fluorescence signals in tumor tissues compared to other non-cancerous tissues in mice treated with LLC2B-L-Cy5.5 (Figure 3A). We then tested the intensity of the fluorescence signal of the tumor tissue in Figure 3C (left) and kidney on the right under the fluorescent confocal microscope. The results show that LLC2B-L-Cy5.5’s fluorescence signal was mainly observed in the tumor, while little can be observed in the kidney. However, the fluorescence signal in the control group with S-LLC2B-Cy5.5 was mainly observed in the kidney instead of the tumor, which means that the scrambled peptide accumulated in the kidney for excretion. Similar results were observed in the KYSE180 xenografted mouse model (Figure 3B). Therefore, we concluded that LLC2B-L-Cy5.5 could specifically recognize tumors in the MCF-7 and KYSE180 xenografted mouse model.

Subsequently, we performed immunofluorescent analysis on selected paraffin sections of human esophageal cancer and tissues surrounding the esophagus cancer to test LLC2B-L-Cy5.5’s targeted recognition ability. The results show that the fluorescence signal of LLC2B-L-Cy5.5 was much higher in tumor tissue than normal tissue (Appendix A), so we confirmed that LLC2B-Cy5.5 can directly target clinical tumor samples.

### 3.4. Peptide Stability Analysis In Vitro

Furthermore, we investigated the stability of LLC2B. By using high-performance liquid chromatography (HPLC), we characterized the map of LLC2B and analyzed the degradation of LLC2B at room temperature at different time points in sterile water. As shown in Figure 4A, we can observe that LLC2B did not degrade at room temperature over 142 h.

Then, we incubated LLC2B with human plasma at 37 °C for 144 h. In the incubation period, LLC2B degradation was analyzed at different time points by using HPLC. The results show that LLC2B had little degradation at 37 °C over 144 h and maintained 95% stability at the last checkpoint. This means that LLC2B is highly stable in human plasma against degradation by proteases and peptidases (Figure 4B).

### 3.5. LLC2B Conjugation with DM1 Inhibits the Proliferation of MCF-7 Cells

Since LLC2B could specifically bind to FGFR2-positive cells and was highly stable in human plasma, we then made PDCs of LLC2B conjugated with DM1 to assess their cytotoxic effects on tumor cells. We obtained two types of PDCs by using two different linkers—(3-(2-pyridyldithio) propionic acid and 3-maleimidopropionic acid)—coupled with the toxin DM1 (Figure 5A,B) and named them LLC2B-SS-DM1 and LLC2B-Mal-DM1. The cytotoxic effects of both PDCs on the tumor cells were assessed and compared in a clone-formation assay. The results show that DM1 could completely suppress the clone formation of MCF-7 cells at 0.001 µg/mL, while LLC2B-SS-DM1 could completely suppress the clone formation of MCF-7 cells at 0.1 µg DM1 equiv./mL. A concentration of 1 µg DM1 equiv./mL was required for LLC2B-Mal-DM1 to achieve the same suppression effect. The control, LLC2B, barely suppressed the formation of cell clones (Figure 5C,D). Moreover, we perform the LLC2B-Mal-DM1 and LLC2B-SS-DM1 stability assay in murine plasma by LC-MS. The results showed that LLC2B-Mal-DM1 was more stable than LLC2B-SS-DM1 in murine plasma (Appendix A) while lower than LLC2B. It is suggested that the linker played key role in the stability of LLC2B-SS-DM1 and LLC2B-MAL-DM1.

### 3.6. Antitumor Efficacy of LLC2B Conjugated with DM1 In Vivo

We investigated the antitumor efficacy of those two PDCs on xenografted mice in vivo. LLC2B-Mal-DM1 at 1.0, 2.0 and 4.0 mg DM1 equiv./kg per week was injected into individual mice according to the tolerated concentrations of DM1 in mice: 1.0–1.5 mg/kg once per week. However, LLC2B-SS-DM1 at 0.5, 1.0, 2.0 and 4.0 mg DM1 equiv./kg was injected in the test due to its higher suppressive effect compared to LLC2B-Mal-DM in the clone formation experiment (the drug concentration used was based on the amount of the substance DM1).

As shown in Figure 6A,C, LLC2B-Mal-DM1 at 2.0 and 4.0 mg DM1 equiv./kg and LLC2B-SS-DM1 with 0.5 mg DM1 equiv./kg had clear antitumor effects compared to the DM1 control groups (Data shown in Appendix A). According to the mouse body weights, there were probably no acute life-threatening conditions in any of the groups (Figure 6B,D). Moreover, the comparison of both PDCs indicated that LLC2B-SS-DM1 had a better effect than LLC2B-Mal-DM1 against the tumors.

## 4. Discussion

The OBOC peptide library used in this study was designed on the basis of the core area of QAE for one short peptide that can bind to FGFR2 found by our team. For the OBOC library constructed on a specific motif, it is easier to obtain desired peptides of high affinity for the target protein. For example, Lam and colleagues designed an OBOC library based on the RGD motif, and screened and obtained peptides named LXW7 and LXW64 for αVβ3 integrins with high specificity. In our study, the selected LLC2B could specifically bind the FGFR2 protein (Figure 1A). It also had a high binding affinity for FGFR2-positive cancer cells in vitro and in vivo (Figure 2 and Figure 3), which indicates the effectiveness of this specific peptide library for screening.

In this study, we also investigated the stability of LLC2B. The results show that it was very stable not only in sterile water but also in plasma. After 144 h, it retained over 90% sequence integrity (Figure 4). However, in the in vivo assays, we observed that, 24 h after tail vein injection, LLC2B was found to accumulate in the kidneys, except for in tumor tissue in the test group. We concluded that LLC2B had been gradually discharged through metabolic circulation by the body’s urinary system since LLC2B has a low molecular weight. Some of the LLC2B specifically targeted the receptors on the membranes of the tumor cells. Thus, we could observe some extra LLC2B remaining in the kidney after 24 h. At the same time, because LLC2B contains unnatural amino acids, the protease could not recognize and degrade it quickly. The feature also ensures the efficiency of LLC2B in identifying and targeting tumor cells in vivo. It has shown its potential in tumor diagnosis and targeted drug delivery.

Maytansine and its derivatives (maytansinoids DM1 and DM4) are cytotoxins structurally similar to geldanamycin, rifamycin and ansatrienin. They are highly potent inhibitors of microtubule assembly for mitotic arrest and kill tumor cells at subnanomolar concentrations [23,24,25,26]. However, maytansine failed to be approved as an antitumor agent after clinical trials due to its lack of tumor specificity and unacceptable systemic toxicity [27]. However, in 2013, an FDA-approved ADC named Kadcyla used DM1 as a cytotoxin to conjugate to the antibody trastuzumab for the treatment of HER2-positive breast cancer. Because of Kadcyla’s good clinical results, we used DM1 as a toxin to similarly construct a PDC in our study. As expected, lower toxicity of DM1 was observed upon conjugating the target ligand LLC2B. Our results prove that LLC2B-SS-DM1 and LLC2B-Mal-DM1 have suppressive effects on the proliferation of FGFR2-positive cells in vitro and in vivo without noticeable toxicity (Figure 5 and Figure 6).

In this study, PDCs formed by coupling LLC2B with the toxin DM1 using different linking molecules (3-(2-pyridyldithio) propionic acid and 3-maleimidopropionic acid) had different tumor-inhibiting effects. Our results show that LLC2B-SS-DM1 was better than LLC2B-Mal-DM1 at inhibiting the proliferation of cancer cells (Figure 5C,D and Figure 6A,C). This indicates that linker molecules play an important role in the antitumor activity of peptide conjugates. We assume that, after the linker was coupled to DM1, it had some effects on the molecular structure of DM1, which would affect its cell toxicity. Another assumption is that different linker-conjugated toxins have different stabilities, resulting in different leakage effects of DM1, affecting the antitumor effects of PDCs. Therefore, the selection of an appropriate linker is very important for the application of peptide-conjugated drugs. As the toxicity of PDCs in vivo is a major concern for further clinical development, further safety research is necessary to determine which one is more suitable for clinical research between LLC2B-SS-DM1 and LLC2B-Mal-DM1.

## 5. Conclusions

We screened a ligand peptide for FGFR2 based on the OBOC specific peptide library and named it LLC2B. We also confirmed that LLC2B has the ability to recognize the FGFR2 protein and FGFR2-positive tumor cells and tissue in vitro and in vivo. Meanwhile, we verified that LLC2B has promising chemical stability in vitro and in vivo.

Then, we coupled two types of PDCs with the toxin DM1 by using different linkers. We also confirmed that those two PDCs can suppress tumor growth and greatly reduce the toxicity of DM1 at the same time. The results indicate that the two PDCs would be good potential drug candidates for treating FGFR2-positive tumors.

## Figures and Tables

**Figure 1 biomedicines-09-00849-f001:**
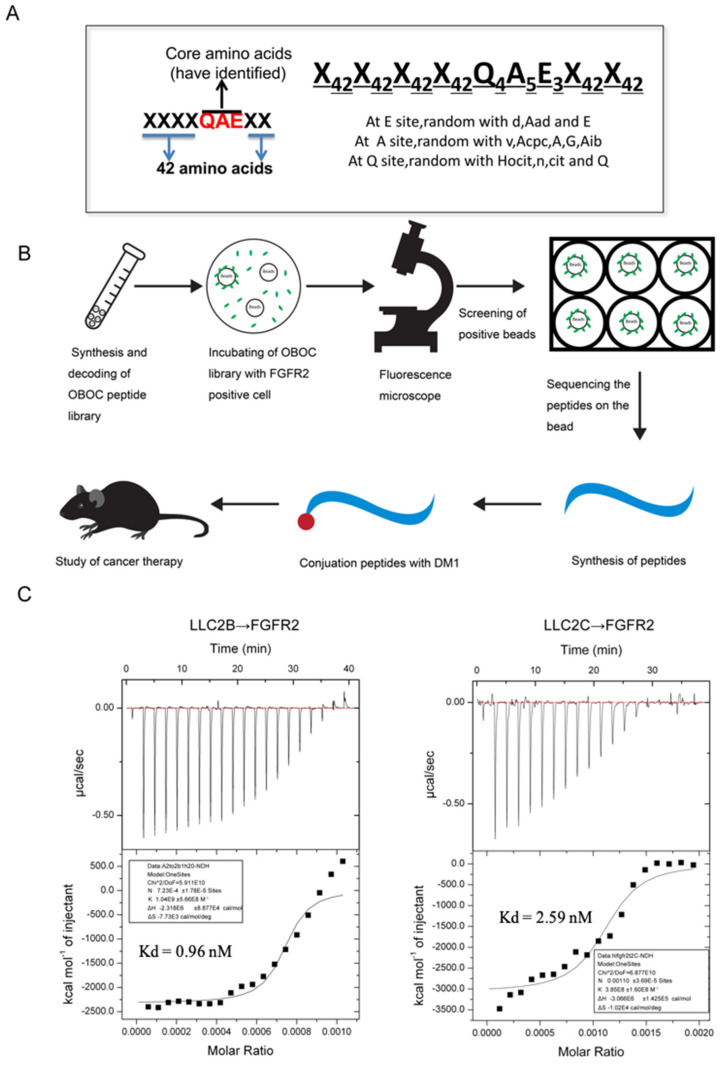
The specific OBOC peptide library, and the affinity of screened peptides for FGFR2. (**A**) Schematic representation of the specific OBOC peptide library. X42 in Figure 1A means any amino acid site of 42 non-essential amino acids. The “QAE” sequence is the core sites of the one antitumor short peptide, E3 means the amino acids of this site could be any amino acids of d, Aad and E; A5 means the amino acids of this site could be any amino acids of v, Acpc, A, G and Aib; Q4 means the amino acids of this site could be any amino acids of Hocit, n, cit and Q. (**B**) Schematic representation of peptide screening and application. (**C**) The affinity of screened peptides LLC2B and LLC2C for binding to FGFR2 according to ITC analysis. LLC2B to FGFR2: the dissociation constant is 1.04 × 10^9^ M^−1^, Kd = 0.96 nM; LLC2C to FGFR2: the dissociation constant is 3.85 × 10^8^ M^−1^, Kd = 2.59 nM. All experiments were repeated 3 times, and each error bar represents SD.

**Figure 2 biomedicines-09-00849-f002:**
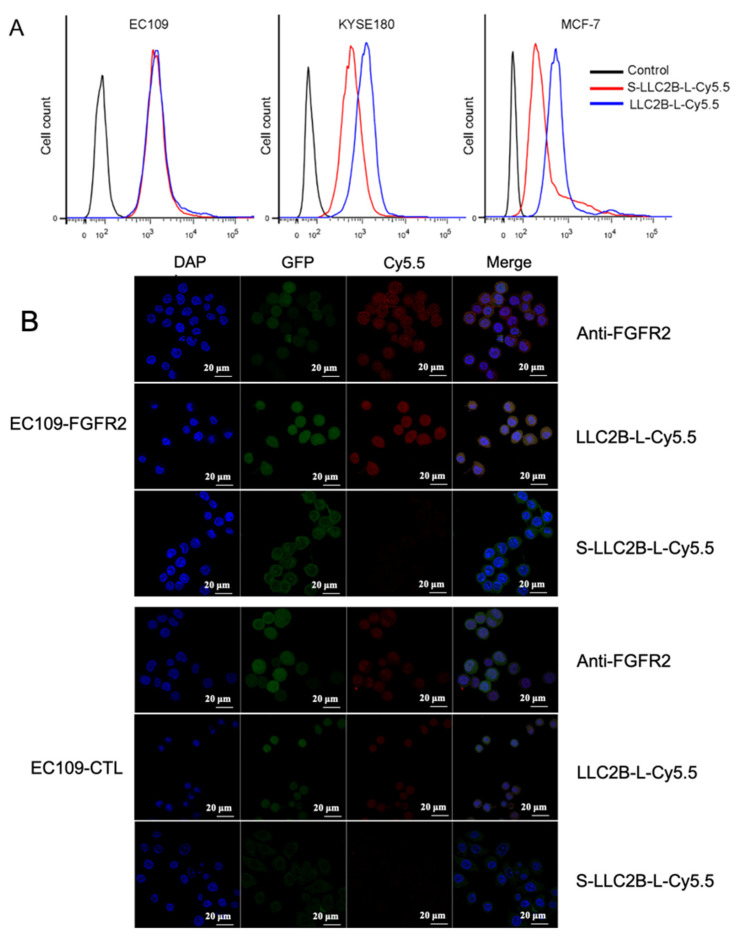
FGFR2-positive tumor targeting of LLC2B in vitro. (**A**) Binding to target cells of LLC2B-L-Cy5.5 detected by flow cytometry in EC109, KYSE180 and MCF-7 cells. LLC2B-L-Cy5.5 or S-LLC2B-L-Cy5.5 were incubated with the cells on ice for 30 min. The results show that, in KYSE180 and MCF-7 cells (FGFR2-positive cells), the fluorescence signal was stronger in the LLC2B-L-Cy5.5 group. However, there was no difference in EC109 cells (FGFR2-negative cells). (**B**) FGFR2 and EC109-CTL cells after incubation with equal amounts of anti-FGFR2, LLC2B-L-Cy5.5 and S-LLC2B-L-Cy5.5 at 4 °C overnight. However, the uptake of LLC2B-L-Cy5.5 in EC109-CTL cells could not be enhanced. The meaning of “GFP” in the column: GFP, which served as the cell indicator, was overexpressed in EC109-FGFR2 and EC109-CTL cells. “Control” means EC109-FGFR2, a recombinant cell line screened using a recombinant lentivirus system expressing both FGFR2 and GFP. EC109-CTL refers to the recombinant cell line screened using the recombinant lentivirus system expressing only GFP.

**Figure 3 biomedicines-09-00849-f003:**
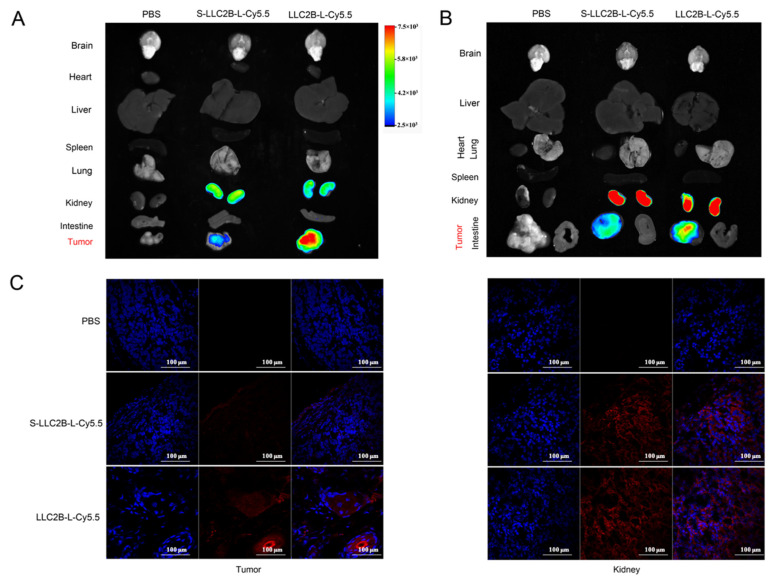
Verification of LLC2B’s affinity in vivo. (**A**) Ex vivo fluorescence imaging of major organs from tumor xenografted mice. Breast cancer MCF-7 tumor xenografted nude mice were monitored by the fluorescence signals with injection of 40 nmol of LLC2B-L-Cy5.5 or S-LLC2B-L-Cy5.5 through the tail vein after 24 h. Mice in the control group received PBS injections (iv). (**B**) Ex vivo fluorescence imaging of major organs from KYSE180 cell xenografted mice. Mice in the control group received PBS injections (iv). (**C**) Observation of frozen tissue section of tumor (left) and frozen tissue section of kidney (right) in Figure A by confocal microscopy.

**Figure 4 biomedicines-09-00849-f004:**
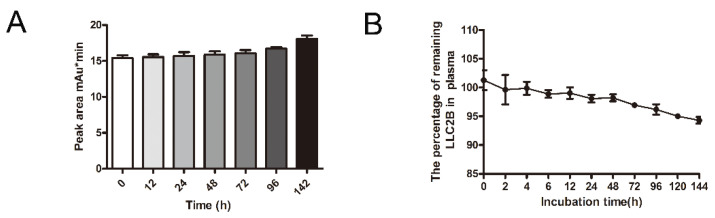
The analysis of LLC2B’s stability in vitro. (**A**) The quantitative results for undegraded LLC2B left in room temperature water for different periods of time. All experiments were repeated three times. (**B**) The remaining percentage of LLC2B after mixing with human plasma and then incubating for different periods of time. All experiments were repeated three times.

**Figure 5 biomedicines-09-00849-f005:**
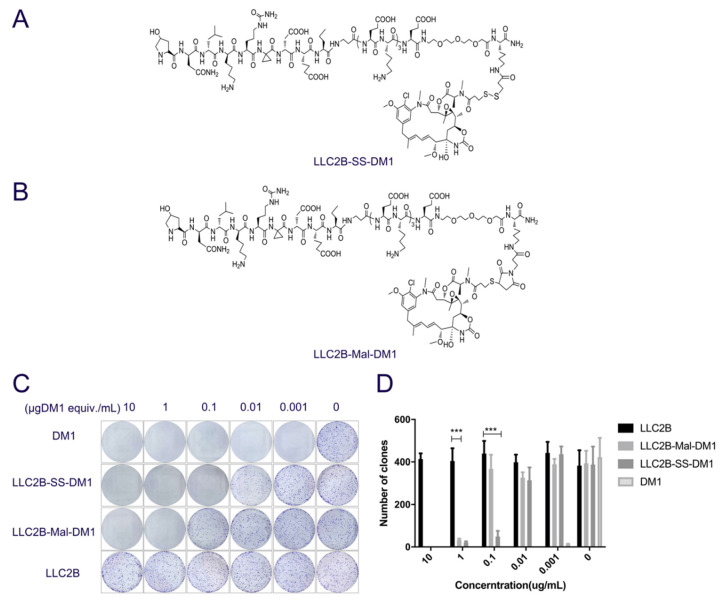
LLC2B conjugation with DM1 inhibits the proliferation of MCF-7 cells. (**A**,**B**) Structure of LLC2B-SS-DM1 and LLC2B-Mal-DM1. (**C**,**D**) The effect of LLC2B-L-DM1 on MCF-7 cells’ clone formation. The effect of two different PDCs on MCF-7 cells’ clone formation, DM1 was the positive control, and LLC2B was the negative control. The counting result for clone formation is in **C**, *** *p* < 0.001. Each error bar represents SD; all experiments were performed three times.

**Figure 6 biomedicines-09-00849-f006:**
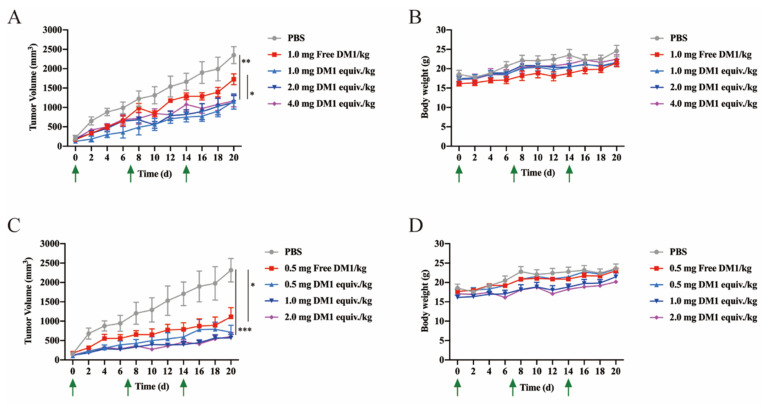
The antitumor efficacy of LLC2B conjugated with DM1 in vivo. (**A**) DM1 was the positive control, PBS was the blank control, the other groups were the different concentrations of LLC2B-Mal-DM1, the arrow indicates the administration time, and the curve shows the effects of different administration methods on the tumor volume of MCF-7. ** *p* < 0.01; each error bar represents SEM. (**B**) The body weight values were measured during the administration depicted in (**A**). Each error bar represents SD. (**C**) DM1 was the positive control, PBS was the blank control, the other groups were the different concentrations of LLC2B-SS-DM1, the arrow indicates the administration time, and the curve shows the effects of different administration methods on the tumor volume of MCF-7. * *p* < 0.05 and *** *p* < 0.001; each error bar represents SEM. (**D**) The body weight values were measured during the administration depicted in (**C**). Each error bar represents SD.

## Data Availability

The data are included in the main text.

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
