# Peer review of "Development and Characterization of a Novel Peptide—Drug Conjugate with DM1 for Treatment of FGFR2-Positive Tumors"

_biomedicines, 2021, doi:10.3390/biomedicines9080849_

Round 1

Reviewer 1 Report

Targeted tumor therapies are more and more important in the treatment of cancer and new powerful ligand-drug conjugates are of special interest.  In their study, the authors comprehensively investigated the efficacy of a novel peptide-drug conjugate with DM1 for treatment of FGFR2-positive tumors in cell culture models and in vivo in a mouse xenograft model.

The general design of the experimental strategy is convincing and sufficient; however, presentation and interpretation of the results contain weaknesses and not all conclusions can be derived from the results.  The literature was not fully addressed, and the manuscript and figures partly lack comprehensibility.  The language should be improved.

Major points:

  1. The entire text contains grammatical and stylistic deficiencies and should be improved, to name just a few of many: introduction: “FGFR dimers and self phosphorylates activate downstream signaling pathways …”, “… such as its small in size …”; results: “The peptide library is consists of …”, “… previous study were used 14”.  The legend of Fig. 1 contains a cross-reference to Table 2, which does not exist, Fig. 3D is identical to the supplementary figure and contains an additional panel (E), which is not explained.  A fundamental revision of the entire text to this end is necessary.

  1. Literature should be considered more adequately, e.g., introduction “Among them, 7 drugs are in clinical phase III” >> provide source; “It plays a role in the pmol and nmol concentration and should not have immunogenicity after entering the human body” >> provide source, discussion: known anti-FGFR2 drug conjugates should be discussed, e.g., Wickstroem K (PMID: 31255687), Sommer A (PMID: 27543601), Borek A (PMID: 29420662).

  1. All exponents in the text are not superscripted, e.g., “… at a cell density of 1 × 103 cells per well …”.  This must be carefully revised in the whole document.

  1. It is written (section 2.12) that the “mice were divided into 9 groups”.  It is unclear whether this occurred randomly or whether a stratified distribution with equal average tumor size and equal standard deviation of tumor size in each group was used.  In the first case, average tumor size and standard deviation must be given for each group and tumor growth must be normalized (see comment to Fig. 6A), in the latter case, the stratified randomization must be described.

  1. Fig. 2: Although the signal on EC109 is identical for S-LLC2B and LLC2B, there is a clear binding signal of LLC2B, which is even higher than that for MCF-7 cells.This is not addressed.  “Control” is not explained, and it is unclear what is shown in the column named “GFP” and what is the meaning of this column.  In the legend it must read “FGFR2-positive cells” (and not “HER2”).  It is unclear what “Repre-Scheme 109” should mean.  Last two sentences: What is shown here is binding and not uptake (there is no uptake at 4 °C).

  1. Section 3.3: The conclusion “has been metabolized easily” is not correct.  The results only provide information on distribution, not on metabolization.

  1. Section 3.5: In the heading, the text, and the figure legend the authors write “antitumor efficacy/treatment”, but these are in vitro experiments, thus there is no tumor present.These experiments address tumor cells.  This should be clearly distinguished from in vivo experiments that are directed against tumors. 

  1. Section 3.6: The conclusion that “there were no significant side effects in all groups” cannot be drawn from body weight observation.  Testing of tissue sections, blood parameters such as transaminases, and complete blood count are required for such a conclusion.  No loss of body weight only indicates that there are probably no acute life-threatening conditions.

  1. Section 3.6: The authors conclude that “the comparison of both PDCs indicate that LLC2B-SS-DM1 has better effect than LLC2B-Mal-DM1 against tumor”.  In contrast to LLC2B, they did not investigate the stability of the conjugate.  Fig. 6C shows a clear effect of free DM1.  Cleavage of the disulfide bridge can result in free DM1, which is then responsible for the effect.  They mentioned this in their discussion, but they did not conduct any experiment with regard to DM1 release.

  1. Fig. 6: The authors compare absolute tumor volume (LLC2B-Mal-DM1) with relative tumor growth (normalized to 1 at start of the experiment) for LLC2B-SS-DM1.  The same is true for body weight, but vice versa.  All experiments should either be shown absolute or normalized for comparison.  If the mice were not distributed to the groups by stratified randomization (i.e., the groups will have different tumor volumes at the beginning of treatment, which looks true for 1.0 mg DM1 equivalent in Fig. 6A for LLC2B-Mal-DM1 starting with a much lower tumor volume), normalization is required for all figures.

Minor points: 

  1. Author statement: “Both authors contribute equally to this work.”  Three authors are indicated with the hashtag sign.

  1. Introduction, first paragraph: “Lung” >> “lung”.

  1. Section 2.13: The third sentence is not complete.

  1. The names of used non-natural amino acids should be provided in full.

  1. Section 3.2: “GFR2” >> “FGFR2”

  1. Section 3.4: “litter degradation” >> “little degradation”

  1. Section 3.4: “of protease” >> “by proteases” (or even better “by proteases and peptidases”)

  1. Section 3.5 and discussion: “SSeimidopropionic acid” >> “3-(2-Pyridyldithio) propionic acid”

  1. Fig. 5D, y-axis: “clooning” >> “clones”

  1. Type always a space between number and unit.

Reviewer 2 Report

This manuscript describes the identification of novel non-natural amino acid peptides targeting FGFR2 which were named LLC2A to LLC2F. LLC2B was subsequently conjugated to the fluorochrome Cy5.5 and characterized in vitro and in vivo for cellular antigen targeting, for human plasma stability and for antitumor activity upon conjugation with mertansine (DM1).

This work could be of potential interest in the pharmacological field of PDCs but, unfortunately, it presents several inconsistencies in the methods and weaknesses in the experimental plan:

- Page 2 (Introduction): there is an inaccuracy as to the number of FDA-approved ADCs to date: this reviewer notes that they are 10, not 8.

- Page 4 (2.9.): a cell viability assay using the CCK-8 kit after 48 hours treatment is described in Materials and Methods, but there is no trace of this type of experiment in the results (only a colony formation assay at 1 week treatment is shown in figure 5). The paragraph 2.9. should be removed.

- Page 5 (3.1.) and Fig. 1: it is not clear how the peptide library is generated, which is the composition of the aminoacidic sequence (Fig. 1A) and how the peptide screening is performed (Fig. 1B). This procedure should be described more clearly and in greater detail; abbreviations and acronyms must always be made explicit in the text.

- Page 7 (3.2.) and Fig. 2A: it is stated that the cell binding experiments by flow cytometry were conducted at 24 hours of treatment, which is too long for this type of analysis (there is a probability of internalization of the target, of cytotoxicity or any other alteration of the system which can confuse the data). On the other hand, in the legend of figure 2 and in the Materials and Methods (paragraph 2.11.) it is stated that the experiment was conducted at a treatment time of 30 minutes on ice (more plausible). What protocol was actually followed? This discrepancy needs to be addressed.

- Page 8 (Fig. 2B): it is not clear how this microscopic analysis experiment was conducted: first, the images are of very poor quality (and without a dimensional bar, as for the images in Fig. 3) although they should have been acquired with a Zeiss LSM700 confocal microscope, as reported in paragraph 2.6. In particular, a staining pattern attributable to membrane targeting is never observed: there is no clear localization of the Cy5.5 signal in the membrane. Furthermore, no comment is made about a possible internalization of LLC2B-L-Cy5.5: what happens to this peptide? If it is not internalized together with FGFR2, how do you explain the targeted activity of the conjugate with DM1? It is essential to investigate whether these PDCs are internalized by endocytosis after binding to the target and, possibly, to measure the kinetics of internalization.

- The caption of Figure 2B must be corrected: “B. Repre[1]Scheme 109. FGFR2, EC109-CTL cells…”

- It is not clear if the cells of the experiment shown in figure 2B were alive or fixed with 4% paraformaldehyde when exposed to the anti-FGFR2 antibody or LLC2B-L-Cy5.5. This must be specified.

- Page 9, Fig. 3: dimensional bars must be added to the figures.

- Page 10 (3.4.): plasma stability experiments should include murine plasma and should also be extended to LLC2B-SS-DM1 and LLC2B-Mal-DM1: these data are essential to have a reference for the in vivo efficacy experiments shown subsequently.

- Page 11, Fig. 5C and D: the antiproliferative effect of LLC2B-SS-DM1 and LLC2B-Mal-DM1 was tested in parallel with DM1 on MCF7 cell colony formation, after treatment for one week. Unfortunately, these data do not demonstrate a targeted activity for the two PDCs: to correlate their cytotoxicity to FGFR2 expression it is necessary to repeat the same experiment with a FGFR2-negative control cell line (such as EC109) or with a non-targeting PDC (for example, S-LLC2B-DM1).

- Page 12, Fig. 6: in vivo efficacy experiments should be supported by pharmacokinetic data, which were not performed or not shown. Furthermore, a negative control (e.g., S-LLC2B-DM1) was not included to demonstrate true targeted activity in vivo. It is strongly recommended that this data be included in any revised version of this manuscript.

Round 2

Reviewer 1 Report

The authors extensively revised the manuscript, and it now reads much better; however, there are still some passages that need to be improved.  In particular, the authors now explained some important issues in their letter to the reviewer, but they did not include this information in the manuscript.

Specific points:

  1. It is written (section 2.12) that the “mice were divided into 9 groups”.The authors now explained it in their response letter (“According to the mean tumor size and standard deviation of each group, a standardized analysis of tumor growth was conducted”) but did not provide the relevant information in the manuscript.  For instance, the authors should provide a supplementary table where they list mean tumor size and standard deviation for each of the nine groups.  This allows the reader to track the results.

  1. Fig. 2: The authors explained in their response letter what “control” and “GFP” mean in their experiment, but they did not include this information in the legend of Fig. 2. Thus, the explanations in their “Response 5”, no. 1 to 4, should be included in the text and figure legend so that the reader can understand the presented results.  Moreover, in the legend of Fig. 2, there is still written “HER2” although this must be “FGFR2”, and the authors stated that they corrected this what is not the case.

  1. Section 3.6: The authors explained why they did not investigate the stability of the conjugate with regard to DM1 release.The argumentation is comprehensible but does not solve the problem.  Fig. 6C shows a clear effect of free DM1.  Cleavage of the disulfide bridge can result in free DM1, which is then responsible for the effect.  For instance, the stability can be measured in isolated blood ex vivo.  If the authors do not have the possibility to sensitively detect free DM1, they should take advantage of the help of others.  Free DM1 is the most critical issue for interpretation and must be experimentally taken into consideration.

  1. Type always a space between number and unit. This is not solved, e.g., section 2.3.  Moreover, it should be written “one hour” or “1 h” but not 1 hour (same for 1.5–2 hours).  Use “µl” instead of “ul”.

Reviewer 2 Report

The points raised were addressed and the manuscript can be suitable for publication in the present form.

Author Response

Thanks for your kind consideration

Round 3

Reviewer 1 Report

The authors now addressed all critical issues.